# Radiometric Assessment of a UAV-Based Push-Broom Hyperspectral Camera

**DOI:** 10.3390/s19214699

**Published:** 2019-10-29

**Authors:** M. Alejandra P. Barreto, Kasper Johansen, Yoseline Angel, Matthew F. McCabe

**Affiliations:** Water Desalination and Reuse Center, King Abdullah University of Science and Technology, Al Jazri Building West, Thuwal 23955-6900, Saudi Arabia; kasper.johansen@kaust.edu.sa (K.J.); yoseline.angellopez@kaust.edu.sa (Y.A.); matthew.mccabe@kaust.edu.sa (M.F.M.)

**Keywords:** hyperspectral, radiometric, UAV, remote sensing, Headwall Nano-Hyperspec, spectroradiometer

## Abstract

The use of unmanned aerial vehicles (UAVs) for Earth and environmental sensing has increased significantly in recent years. This is particularly true for multi- and hyperspectral sensing, with a variety of both push-broom and snap-shot systems becoming available. However, information on their radiometric performance and stability over time is often lacking. The authors propose the use of a general protocol for sensor evaluation to characterize the data retrieval and radiometric performance of push-broom hyperspectral cameras, and illustrate the workflow with the Nano-Hyperspec (Headwall Photonics, Boston USA) sensor. The objectives of this analysis were to: (1) assess dark current and white reference consistency, both temporally and spatially; (2) evaluate spectral fidelity; and (3) determine the relationship between sensor-recorded radiance and spectroradiometer-derived reflectance. Both the laboratory-based dark current and white reference evaluations showed an insignificant increase over time (<2%) across spatial pixels and spectral bands for >99.5% of pixel–waveband combinations. Using a mercury/argon (Hg/Ar) lamp, the hyperspectral wavelength bands exhibited a slight shift of 1-3 nm against 29 Hg/Ar wavelength emission lines. The relationship between the Nano-Hyperspec radiance values and spectroradiometer-derived reflectance was found to be highly linear for all spectral bands. The developed protocol for assessing UAV-based radiometric performance of hyperspectral push-broom sensors showed that the Nano-Hyperspec data were both time-stable and spectrally sound.

## 1. Introduction

Hyperspectral imagers enable the collection of a series of contiguous, very narrow spectral bands, providing a near-continuous spectrum of an object, commonly referred to as the spectral signature. The technique offers several advantages over more traditional multispectral sensors, where few, relatively wide bands are implemented [1,2,3]. While multispectral sensors usually consist of 5–12 bands [2], hyperspectral sensors can comprise hundreds of bands, with a bandwidth generally in the range of 5–20 nm [2]. The benefits of quantifying the spectral response of an object or a material have been well recognized, with applications across disciplines spanning plant sciences for plant pigment, biochemistry and species assessment [4,5,6,7,8], geology for exploration of minerals [9,10], and marine sciences for water quality assessment and benthic community mapping [11,12,13,14,15], amongst many others.

The utility of hyperspectral sensors has been advanced through relatively recent developments in the use of unmanned aerial vehicles (UAVs) as an observation platform. With developments in sensor miniaturization, power supply stability, communication, and storage requirements, the use of UAVs for hyperspectral remote sensing has become a feasible option for acquiring ultra-high spatial resolution hyperspectral data [8]. UAV-based hyperspectral sensors can be classified broadly into several groups. Following Aasen et al. [16], these include point-based spectrometers, spectral 2D imagers, and push-broom sensors. Point-based spectrometers or calibrated spectroradiometers provide a high spectral resolution, dynamic range, and signal-to-noise ratio. However, their data generally have no spatial reference [16] and, being point-based, cannot be displayed as a raster image. Spectral 2D imagers constitute the second group of hyperspectral sensors. They can record all the bands sequentially or simultaneously. Among the spectral 2D imagers, image-frame sensors are those that record the bands sequentially. These enable data acquisition of high spatial resolution and often allow a user to pre-select the desired bands [16]. However, post-processing is required to correct offsets of the bands within individual cubes [17,18,19]. Snapshot 2D imagers comprise the second type of spectral 2D imagers. Their main advantage is that they record all the bands at the same time, thus avoiding the need for spatial co-registration [20]. On the other hand, push-broom sensors capture one line per exposure (1D), forming one image line after the other. The spectral signature of objects within the instantaneous field of view (IFOV) of the camera is contained within each pixel [16]. UAV-mounted push-broom cameras allow the collection of very high spatial (and temporal) resolution data. However, additional equipment such as Global Navigation System Satellite/Inertial Measurement Unit (GNSS/IMU) sensors and data logging units is needed for georeferencing the images, since the data collection process of a push-broom sensor is sensitive to flight dynamics and orthorectification can be a challenging procedure [16].

Undoubtedly, UAV-based hyperspectral remote sensing offers a convenient alternative over traditional space and airborne platforms, particularly for small area mapping and frequent monitoring applications, providing both high spatial and temporal resolution on demand [21,22]. However, as with any advanced technology, the emerging nature of their application demands an assessment of both capabilities and limitations [2,14]. Among the advantages is the recording of hundreds of very narrow and continuous bands [2], providing greater spectral detail, which facilitates improved spectral discrimination [2,14,23]. Spectral libraries can then be built from the collection of different spectra. Hyperspectral remote sensing also facilitates the construction of spatial models with detailed spectral characteristics suitable for enhancing classification [14]. Of course, these advantages also come with some potential drawbacks. Challenges of hyperspectral remote sensing are often related to interrogating the high-dimensional data and subsequent processing [2,14], geometric and radiometric correction, data quality assurance, and also discriminating relevant information from vast datasets [2].

Sensors record data at a specific number of spectral bands and are affected to various extents by external factors such as varying illumination conditions or viewing geometry [21], flying height, geometric distortion and orthorectification issues [24,25], as well as by internal factors such as sensor type and sensitivity, the spectral response function and dark current levels. Internal factors can be accounted for during sensor calibration, which is a critical step in the fabrication and deployment of a hyperspectral camera [2,26,27,28] and is needed for establishing standardized and objective measurements. For example, so-called ‘dark current’ noise levels, which represent the relatively small electric current flowing in a photoelectric device that is not exposed to any incident illumination, are higher in complementary metal-oxide semiconductors (CMOS) than coupled charged devices (CCD) due to the inherent functioning of these devices [2,29]. Thus, dark current correction and standardization guarantee that measurements are accurate, credible, reproducible, and comparable between different experiments [9].

Sensor-related calibration comprises the spectral calibration, the relative radiometric calibration, and the absolute radiometric calibration [16]. Spectral calibration ensures an accurate spectral response of the wavebands (spectral fidelity), while the relative radiometric calibration ensures a uniform output across pixels and time. Finally, the absolute radiometric calibration determines the conversion from digital numbers (DN) to radiance or reflectance units. This is generally referred to as linearity because the relationship between DN and radiance is often linear. Scene reflectance generation can be achieved e.g., using the empirical line method [16], which is a calibration procedure to derive surface reflectance from at-sensor radiance or raw DN [30]. Although the relationship between at-sensor radiance and surface reflectance is also assumed to be linear [30,31], Wang and Myint [30] pointed out the difficulties of obtaining radiance for some imagery, as well as the problems often arising from the characteristics of the calibration panels used for the method. Hence, they further investigated the relationship between DN and surface reflectance. Conversion into normalized radiance or reflectance units allows hyperspectral data across different experiments and between different sensors, locations, times, and conditions to be compared.

Hakala et al. [32] designed a calibration procedure for a 2D format tuneable Fabry–Pérot interferometer-based hyperspectral camera coupled with a system for direct reflectance measurements from a UAV. Both a spectral and an absolute radiometric calibration were performed. Their stability evaluation, mimicking a drone campaign, consisted of taking radiance measurements at 5 and 30 min, and if the relative radiance difference between both times was above 5%, they considered the band to be unstable. After a spectral calibration (performed at the National Physical Laboratory (NPL) in Teddington, UK), they found that the spectral response parameters (peak center wavelength and full width at half maximum (FWHM)) differed from the manufacturer’s reported parameters. Aasen et al. [20] and Yang et al. [33] both worked with the CubertUHD Firefly snapshot hyperspectral camera. Yang et al. [33] examined the linearity between DN and radiances. Their results showed a linear relationship between DN and radiance, with the goodness of fit varying from band to band, but with a coefficient of determination (R^2^) above 0.998. Their spectral calibration assessment, performed with a Hg/Ar lamp, showed that the peaks from the CubertUHD camera lagged with respect to the emission lines of the lamp and that the FWHM increased with increasing wavelength. After performing a spectral calibration, a shift of only 1 nm occurred in the wavelength of each band, which was considered negligible given the FWHM. Hence, the wavelengths were considered stable. For the same camera, Aasen et al. [20] evaluated dark current levels within 20 min, which they found to increase and pointed out the presence of dark current spatial inhomogeneities within one band. However, they considered dark current levels to be low enough to have an insignificant effect.

Arngren [34] presented an assessment of a near-infrared (NIR) hyperspectral line scanner manufactured by Headwall Photonics. In this technical note, they found that for the InGaAs sensor, bands recording wavelengths at each extreme of the spectrum had lower sensitivity and a lower signal-to-noise (SNR) ratio. Additionally, they found pixels with consistently larger than average dark current DN values across all bands. Assessment of dead pixels (fixed at zero or at maximum DN counts) showed that the percentage of those was below the acceptable limit set to 0.9% by the camera manufacturer. Lastly, their spectral performance assessment showed shifts between the acquired spectrum and the corrected one against a calibration standard. Liu et al. [35] assessed the radiometric and spectral calibrations of the UAV-mounted visible/near-infrared imaging spectrometer push-broom scanner during a flight campaign. The radiometric calibration produced a linear relationship with high R^2^ coefficients between DN and MODTRAN 5 simulated at-sensor radiances. For the spectral calibration, they used two different methods, with one requiring in situ measured surface reflectance and atmospheric parameters, while the other one was applied without the in situ surface reflectance. Their results showed that both methods produced almost identical results.

The field of UAV-hyperspectral cameras is in constant evolution with new sensors frequently being designed and deployed, and despite being carefully calibrated by manufacturers, hyperspectral cameras of any type may need recalibration due to variations between laboratory and flight conditions [35] or due to small shifts of the spectral parameters over time [33]. It is therefore good practice to assess the performance and reliability of new products as well as conducting periodical assessments of the cameras in use. Usually, most of the assessment and calibration experiments require the use of expensive and specialized instrumentation not readily available in most remote sensing laboratories, where cameras are purchased for end-user applications, and sending the cameras back to the manufacturers for assessment may not be practical. Additionally, data calibration methodologies applied during surveys may be affected by atmospheric and illumination conditions, limiting the generality of results. Overall, there is a distinct lack of standardized protocols for consistent evaluation of the radiometric performance of UAV-based hyperspectral sensors, despite the significant effects that spectrally inconsistent data may have on mapping applications.

In this study, we outline and implement an approach for assessing the radiometric performance of miniaturized hyperspectral push-broom sensors, with a goal to ensure that a consistent, reliable, and repeatable collection of hyperspectral data can be achieved. The proposed approach is meant to be implemented with the aid of instruments commonly found in most remote sensing laboratories and to be undertaken in a relatively short period and without requiring extensive computing resources. As an example application, the performance of a Headwall Nano-Hyperspec imaging sensor was examined for its ability to provide accurate and reliable hyperspectral retrievals. To do this, we undertook a series of evaluation steps. First, the DN counts and radiance values of the dark current, a white reference, and their associated anomalies over time and across pixels were collected to characterize the sensor’s stability and precision. A second assessment involved an examination of the spectral wavelength location of the sensor, utilizing a Hg/Ar light source and a field spectroradiometer. Finally, the conversion from radiance to reflectance through the empirical line method was evaluated using different grey-scale radiometric calibration panels and comparing these to spectroradiometer measurements. It should be noted that the developed approach is not a calibration procedure, but rather a workflow for evaluating the radiometric performance of small hyperspectral push-broom sensors suited for UAVs.

While most published experiments for assessing and calibrating UAV-based hyperspectral cameras are focused on snapshot cameras, very few examples were identified for push-broom hyperspectral cameras [34,35]. One of the most extensive assessments of a hyperspectral camera is presented in [34]. Although most of the key points for evaluating the camera’s performance are summarized, it lacks guidance and analysis regarding the temporal stability of the sensor. Additionally, it implements costly specific standards for the NIR region that may not be readily available. Moreover, most of the documented experiments make use of very specialized instruments that might not be acquired by end-user remote sensing laboratories. One of the innovative aspects of this paper is that it is oriented to UAV users, who collect data under the assumption that their cameras are accurately calibrated by the manufacturer, and currently have no options for verifying the radiometric data quality of their push-broom hyperspectral cameras. Our work addresses this gap by providing a simple yet effective protocol for evaluating the data quality and radiometric performance of UAV-based push-broom hyperspectral cameras.

## 2. Materials and Methods

### 2.1. Overview of the Nano-Hyperspec Sensor

The Nano-Hyperspec sensor is one of a number of recent hyperspectral systems that have been specifically designed for UAV-based remote sensing. The sensor, which is based on CMOS technology, is relatively lightweight at 0.52 kg and occupies a rather small footprint with dimensions of 76.2 mm × 76.2 mm × 119.9 mm. The Nano-Hyperspec is a push-broom sensor, i.e., imagery is collected along the direction of flight, line by line, with each line captured at a unique moment in time, corresponding to an instantaneous position and attitude of the UAV. Each line of pixels comprises 640 spatial pixels and 270 spectral bands with a total of 172,800 pixel–waveband combinations. The sensor collects 12-bit data, i.e., 4096 brightness levels, across the visible and near-infrared spectrum, ranging from 400 to 1000 nm, with a sampling interval of 2.2 nm and a FWHM of approximately 6 nm [36].

### 2.2. Performance Assessment Protocol

For the evaluations performed here, the exposure time and frame period, i.e., the inverse of the frame rate, were set to the same value in each experiment, but the value between experiments differed depending on light conditions. Exposure time refers to the time the sensor is exposed to light, while the frame period refers to the time in which an individual frame is collected, with both expressed in milliseconds (ms). The exposure time is set to optimize the instrument’s internal gain settings for best signal-to-noise ratio during the brightest conditions encountered during data collection while preventing the risk of detector saturation, i.e., signal over-range. Here, a frame refers to an array of 640 pixels, i.e., the spatial resolution of the sensor. Frames per cube refer to the number of frames that are saved under one .hdr file, which in this study is set to 1000 for all experiments. Since the experiments were stationary, we averaged the 1000 frames in each file to get a representative value of a time interval within 6 and 12 s, depending on the exposure used. The value of 1000 frames per cube was considered appropriate to capture data of small intervals of time while avoiding excessively big files, which make processing difficult. Table 1 summarizes the main features of the collected data per experiment. Figure 1 summarizes the steps undertaken for the complete assessment as a workflow of the protocol proposed in this study and described in detail in the following sections.

### 2.3. Relative Radiometric Calibration Assessment

Push-broom sensors collect data in successive linear arrays of picture elements. For each pixel in the array, spectral data are collected simultaneously and independently. Small differences between the sensors of each pixel can create a pattern of across-track noise, known as vertical striping [37]. Additionally, there may be a pattern of along-track noise caused by temporal variations induced by dark current effects and the optical system [37]. While this research did not seek to remove these noise patterns, it is important to evaluate the spatial response of the camera across its 640 pixels and to study any temporal variations of these throughout average UAV flight times, i.e., generally 30 min. Spatial pixel measurements should be homogeneous for the same feature, and temporal variations between the first and last measurements should be small enough not to affect the signal significantly, or at least be consistent to allow for simple removal. The relative radiometric performance of the Nano-Hyperspec sensor was assessed in two main steps: (1) a dark current assessment for the area detector, i.e., the mechanism which converts incident photons into electrons to quantify light intensity [30]; and (2) a white reference assessment for the optical system, i.e., the system that forms the actual image and the spectrometer [37]. Initially, raw DN data were examined. Subsequently, DN counts were converted into radiance values using Headwall’s SpectralView software to allow for a more detailed examination of the temporal variability. Calibration files provided by Headwall allowed the conversion from DN to radiance of the Nano-Hyperspec data. The transformation provides measurements in physical units of milliwatts per square centimeter per steradian per micrometer (mW/cm^2^·sr·µm), thus eliminating the problem of specific measuring scales for each sensor. Further details of these two experiments are provided below.

The basic setup for the laboratory experiments consisted of a box-like stand constructed with aluminium bars. A horizontal bar, placed at approximately 80 cm from the floor, was used for placing the light source during the white reference experiments. The camera was placed inside the stand and pointed at nadir. The camera was aligned horizontally using a spirit-level. For all the experiments, a black fabric was used to fully cover the box-housing to minimize external light sources. Stands of different heights were used to place the targets at the desired distance from the camera. For the relative radiometric calibration assessment experiments, the data collection was done in the morning. Subsequently, the camera was left to cool down until the following morning before running a new experiment.

#### 2.3.1. Dark Current Assessment

Dark current can be defined as the residual current that flows through a photo-sensible device when there is no incident radiation upon it [38], due to the random generation of electrons. Dark current is temperature-dependent [39], and its effects increase with increasing temperature. The aim of the experiments was first to establish if dark current levels were uniform across all 640 spatial pixels in an array and to identify ‘hot pixels’, i.e., pixels in DN or radiance values with above-average dark current levels due to saturation [38]. To assess the spatial response, the value for each pixel–waveband combination along all the frames was averaged, and variability was assessed through the standard deviation. Ideally, each of the 640 pixels in the array should show similar readings for each waveband, which would indicate consistent dark current levels. The evaluation was performed for both DN measurements and radiance. Additionally, a careful examination was undertaken to identify hot pixels in the raw data and determine if they could be identified in the radiance images.

A temporal assessment was also undertaken to determine how dark current levels fluctuate during a standard time interval of UAV-based sensor operation (30 min) and whether these variations are significant enough to affect sensor accuracy. Dark current DN levels were recorded for a 30 min period, at exposures of 6 and 12.5 ms: the first value being a common exposure value employed during flight campaigns under sunny conditions, and the latter being the exposure used for the white reference experiments. For the experiments, the camera was turned on, and the image capture initiated (with the lens cap on), within 2 min after power-up. A total of four experiments were run for this purpose: two at 6 ms exposure and two at 12.5 ms exposure, to verify consistency between experiments. Temporal variability was assessed after converting DN into radiance since it is a standard unit that allows for comparison with other sensors. The assessment was done by calculating the relative difference (percentage of change) between the first measurement at time t = 0, and the measured value at each successive minute for the duration of the experiment, i.e., time t = 30 min. This relative difference was calculated for all pixel–waveband combinations to get the minimum and maximum variations during the experiment. For practical purposes, results were averaged, first for all possible pixel–waveband combinations and then for all the pixel–waveband combinations within a specific region of the spectrum: Blue (400–500 nm), Green (500–600 nm), Red (600–700 nm), and NIR (700–1000). Finally, the percentage of change between the first and last radiance measurements was used to quantify pixel–waveband combinations with a net increase or decrease in their radiance values with respect to the first measurement. For the dark current assessment, only the basic setup and the Nano-Hyperspec sensor were required. The lens cap was left on before turning on the camera and the light in the room was turned off before starting the data recording.

#### 2.3.2. White Reference Assessment

The goal of the white reference assessment was to establish whether the spectral signature was consistent across the spatial pixels and over time between frames. Experiments were conducted utilizing a white Spectralon panel (Labsphere Inc., North Sutton, NH, USA) with 99% reflectance across a wavelength range from 250 to 2500 nm. Two repeat experiments, each running for 30 min, enabled evaluation of the DN and radiance fluctuations over time. The two experiments were conducted with exposures of 12.5 ms, i.e., the minimum exposure required to attain optimized DN with the illumination source. Similar to the dark current experiments, the analysis of spatial and temporal variability was undertaken separately. Spatially, all pixels were expected to have similar readings at a specific wavelength. An extra step was taken for the spatial assessment in the white reference experiments whereby the mean radiance was divided by the mean DN, producing what is referred to here as the Spectralon radiance response, following Rozenstein et al. [40], to allow for comparisons with other common experiments. Temporal variation was assessed in the same fashion as described in Section 2.3.1 for the dark current experiment.

For the white reference assessment experiments, only the Spectralon panel and a halogen lamp as light source were required besides the basic setup described in Section 2.3. Special attention was given to the placement of the light source (halogen lamp) in relation to the Spectralon panel and sensor to attain a consistent spectral signature among all pixels and with optimized DN to reduce the signal-to-noise ratio and prevent saturation. The camera angle was kept constant at nadir, and the illumination source angle was also constant at 45° to allow for comparison between the two experiments. The setup was protected from any other incoming light with black fabric, which was especially important for this assessment. For these experiments, both the camera and target were stationary.

### 2.4. Spectral Calibration Assessment

Undertaking a spectral waveband calibration is one of the most critical calibration elements that is conducted when a camera is being manufactured [26]. It guarantees that the defined spectral channels of a camera are recording the associated wavelengths, i.e., the spectral fidelity [41]. For this, calibration lamps from different rare gases or metal vapors are often used to produce narrow, constant, and specific wavelength emission lines [42]. The goal of this experiment was to validate the manufacturer’s spectral waveband calibration. In this case, we assessed if the wavelength emission lines of a Hg/Ar lamp matched those recorded by the Nano-Hyperspec sensor. The HG-1 Hg/Ar lamp (Ocean Optics Inc., Dunedin, FL, USA) had Hg and Ar wavelength emission lines from 253–580 nm and 696–922 nm, respectively. Table 2 presents the emission lines for the HG-1 lamp with the lines specified for Hg and Ar separately. The exposure for the Nano-Hyperspec sensor of 1000 ms was used to attain the desired DN counts, adequate for comparison with spectroradiometer measurements. At exposure times lower than 1000 ms, no clear signal could be recorded. For the measurements with the Nano-Hyperspec sensor, the lamp was held vertically, pointing the light source towards the camera lens (camera at nadir). The distance between the camera lens and the light source was approximately 2 cm. No other light sources were present during the experiment.

An ASD FieldSpec-4 (Analytical Spectral Devices Inc., Boulder, CO, USA) spectroradiometer was also used for this experiment, with its optical fiber cable directly connected to the light source. The readings were taken on the DN scale of the Fieldspec-4 and were not further processed. These measurements were taken as the standard for comparison. The spectrum of the Hg/Ar lamp collected with the spectroradiometer was compared with that measured by the Nano-Hyperspec sensor. These observations were also related to documented Hg and Ar wavelength emission lines. For the spectral calibration assessment, only the Hg/Ar lamp and the spectroradiometer were needed, besides the Nano-Hyperspec.

### 2.5. Conversion from Radiance to Reflectance

We sought to establish the relationship between data acquired by the Nano-Hyperspec sensor converted to at-sensor radiance using the SpectralView software and coincident reflectance data acquired with the FieldSpec-4 spectroradiometer. Obtaining imagery in reflectance units is required for the comparison of different data sets collected for different areas or the same area over time. We chose to perform the empirical line method to obtain regression equations to convert from at-sensor radiance to at-surface reflectance given that radiance values could be derived from Headwall’s software. At-sensor radiance values of the Nano-Hypserspec sensor were related to reflectance using three different types of near-Lambertian radiometric calibration panels, for which Nano-Hyperspec radiance and ASD Fieldspec-4 spectroradiometer reflectance values were obtained. The three different radiometric calibration panels comprised: (1) a Spectralon gray-scale panel with nominal reflectance values of 12%, 25%, 50%, and 90%; (2) seven Masonite (steam-cooked and pressure-molded wood fibers) panels painted with three coats of matte paint, ranging from black to white, following the procedure of Wang and Myint [30] and Johansen et al. [44]; and (3) six oak plywood panels painted similarly to the Masonite panels. Spectralon panels were chosen as they have diffuse reflectance properties which show a nearly unchanged spectral response from 250 to 2500 nm, making them the ideal reflectance standards. Masonite panels were chosen as they are relatively cheap and commonly available, and have been utilized previously [30], showing near-Lambertian properties when coated with matte paint. Oak plywood was chosen as the third material because it is an extremely common and inexpensive, and provides a readily obtainable and replaceable solution when other types of panels may not be available. Oak plywood panels have a smooth surface in contrast to the characteristic texture of the Masonite. The suitability of all three panel types was tested in this experiment.

To replicate similar atmospheric conditions to those in the field, the experiment was conducted outdoors between 9:00 and 10:30 a.m. (local time) under clear sky conditions. White references were collected for both the Nano-Hyperspec sensor and the spectroradiometer before the measurements of each set of calibration panels. Using the empirical line method [30,31] for each set of panels and all 270 bands, a line of best-fit was obtained through linear regression between the reflectance measured with the spectroradiometer and the radiance measured with the Nano-Hyperspec sensor. The fit of this line was evaluated using R^2^ and the residuals for the linear regressions vs. the fitted reflectance values, calculated from the regression equations, for the three different sets of panels.

## 3. Results

### 3.1. Relative Radiometric Calibration Assessment

#### 3.1.1. Dark Current Assessment

An initial analysis of the DN showed that the mean signature across dark current experiments was consistent with values between 110.2 and 128.8 DN for all the pixels and bands (Figure 2a) on a scale from 0 to 4000 DN. The lowest readings were found for the bands around 400 nm. The edges of the pixel array (pixels below 100 and above 500 out of the 640 pixels in an array) showed consistently larger DN values across all the bands, indicating that these have the largest dark current levels. Looking at the spectral bands and the pixels in Figure 2a, it seems there are two superimposing patterns, i.e., one producing vertical strips (across the pixels) and one producing horizontal strips (across the bands). The latter means some bands show larger DN values across all the pixels, and some pixels show larger DN values than their neighbor across all the bands. This behavior ultimately produces a gridded appearance.

Figure 2b shows that the spatial pixels at the edges of the array have the largest variability in their dark current readouts, indicating that in the DN domain these pixels are generally noisier due to dark current effects and hence fluctuations in temperature. The gridded pattern is less evident in Figure 2b, where only the horizontal strips can be recognized, showing groups of pixels with larger or lower variability of the dark current levels. This indicates that variability in dark current levels is more influenced by the pixel position in the array than by the wavelength being recorded. The latter is true except for the wavelengths between 400 and 500 nm, showing the largest variability and forming a yellow vertical strip. In general, the standard deviations in the dark current experiments with 6 ms exposure were around 0.096–0.100 DN for most of the pixels.

Figure 2c shows the dark current levels in the radiance domain. The patterns observed in the DN domain were not evident when analyzing the derived radiance values (Figure 2c). This was attributed to the use of gain and offset parameters in the conversion from DN to radiance values, which generally reduces pixel-to-pixel and band-to-band variations. The lowest radiance values at approximately 0.4 mW/cm^2^·sr·µm were found between 600 and 700 nm. Values close to 400 nm and beyond 800 nm were over 1 mW/cm^2^·sr·µm. Values around 1000 nm were considerably larger at approximately 4 mW/cm^2^·sr·µm. The second experiment with an exposure of 6 ms showed the same pattern. The response was uniform across pixels, which means that dark current levels, measured in radiance units, affect all pixels in the same way or are spatially consistent. The standard deviation was lower, where radiance values were lower and were slightly larger where radiance values were larger. The largest of the standard deviations were below 0.01 mW/cm^2^·sr·µm around the mean radiance, which can be considered low for values of 4 mW/cm^2^·sr·µm in this part of the spectrum.

Similar patterns were observed for the dark current experiment at 12.5 ms of exposure, where lower radiance values were approximately 0.2 mW/cm^2^·sr·µm around 600 nm, and larger values close to 400 nm and beyond 800 nm were between 0.7–2.4 mW/cm^2^·sr·µm. Around 1000 nm, the radiance values reached >2 mW/cm^2^·sr·µm. The experiment performed with an exposure of 12.5 ms also showed that responses were uniform across pixels. Hence, similar patterns but different radiance values were observed using different exposures. These results mean that for different exposure times, the dark current measured in radiance values is different, with lower radiance values for longer exposure times. In this case, the increased exposure reduced the radiance baseline by roughly 50%.

Ideally, radiance measurements of the dark current should be homogeneous across pixels and wavelengths, as they represent “no light” conditions. Variations in the mean dark current signal and its standard deviation in the radiance domain show a consistent response across pixels and are more variable across wavelengths. To remove dark current effects, the dark current signal should be recorded before each survey. Dark current effects and any variability among them should be accounted for by subtracting the dark current spectrum from the measured spectrum. Although it may be implied that a larger exposure results in lower dark current levels in the radiance domain, in reality, the selected exposure is more dependent on the illumination conditions present during a survey.

For the DN responses, 17 pixel–waveband combinations showed values above 135 DN, with only seven pixel–waveband combinations having DN values above 140, which was consistently identified in all four dark current experiments. Six pixel–waveband combinations showing elevated values were found within the blue portion of the spectrum and one “hot pixel” occurred in the NIR part of the spectrum. The hot pixel–waveband combinations were also observable in the radiance experiments, exhibiting values above 1.6 mW/cm^2^·sr·µm in the blue region and 4.9 mW/cm^2^·sr·µm in the NIR region of the spectrum at 6 ms exposure. At 12.5 ms exposure, the radiance values were approximately 0.75 and 2.5 mW/cm^2^·sr·µm for the blue and near-infrared regions, respectively. The seven identified “hot pixels” (pixel–waveband combinations) out of a total of 172,800 combinations would have little impact on the collected data during UAV surveys with the Nano-Hyperspec sensor and, therefore, can be considered negligible for field-derived imagery.

The results of the temporal variability analysis showed that the largest increase and decrease in radiance values were 5.6% and 2.5%, respectively (Figure 3a). Over 80% of the pixel–waveband combinations exhibited a relative difference in radiance values below 2% throughout the 30 min experiment. For the remaining 20% of pixel–waveband combinations, the relative difference ranged from 2.0 to 5.6%. The percentage of pixel–waveband combinations showing an increase larger than 3% was negligible. For these experiments, over 99% of the pixels showed a positive relative difference, which can be attributed to an increase in dark current levels. For the dark current experiments with an exposure of 6 ms, this means that only 750 out of 172,800 pixel–waveband combinations showed a decrease in their last measurement. Additionally, horizontal blue strips in Figure 3a indicate some pixels where the relative difference is preferentially negative.

Figure 3b shows that all wavebands followed the average change in radiance. The average change in radiance showed a slightly oscillating pattern throughout the 30 min, with no evident increase until around 15 min. After 15 min there was a slight increase. The average increase of radiance values after 30 min was below 2%, emphasizing that overall, there was low temporal variability in the radiance measurements of the Nano-Hyperspec sensor, and the response with time was stable.

Increasing the exposure to 12.5 ms resulted in larger relative changes of radiance both above and below the first measurement. The maximum positive (difference between first and last reading) percentage of change was 7.3%. The largest change below the first measurement was 4.1%. This means that increasing the exposure generates a larger variability in the temporal response, i.e., higher and lower values with respect to the first measurements. Approximately 99.5% of all pixel–waveband combinations had differences in radiance values between the first and last measurements <2%, with only 0.5% of the measurements being >2%. The percentage of pixel–waveband combinations showing a difference in radiance values >3% was negligible. For the experiments with 12.5 ms of exposure, 98% of the 172,800 pixel–waveband combinations showed a larger radiance reading in their last measurement, while 2% showed a decrease in their recorded value. The latter means that 2852 pixel–waveband combinations showed a decrease in radiance in their last measurement. All wavebands followed the average change in radiance through time. The average increase in the radiance by the end of the 30 min experiment was <2%. These findings indicate that although the variability is larger for a 12.5 ms exposure, the camera’s response was very similar to that of the 6 ms exposure and can be considered time stable.

The results indicate that dark current levels are different among bands, but are consistent across pixels, which is acceptable for the camera operation. Variability in dark current levels increased with higher exposure. However, the average response can be considered time stable. The normal exposure times of the Nano-Hyperspec sensor during flights range from 3 to 6 ms. Hence, variability in dark current levels from the electronic system alone should be lower, with overall negligible effects on the quality of field-derived imagery. Overall, it can be concluded that no warm-up period is required before camera deployment due to dark current or optical effects.

#### 3.1.2. White Reference Assessment

The spectral signature in DN for the white reference followed the emission pattern characteristic of halogen lamps, with a peak between 600 nm and 800 nm for all 640 pixels (Figure 4a). As expected, the standard deviation was higher where the DN readings were larger, i.e., between 600 nm and 800 nm (Figure 4b). However, the peak standard deviation shifted slightly towards lower wavelengths (around 650–700 nm) of the spectrum with respect to the mean signature’s maximum at about 700 to 750 nm.

Figure 4c corresponds to the mean spectral signature in radiance units. The image shows a gradient, with the lowest values around 400 nm and the largest around 1000 nm, differing from the pattern displayed by the DN. The standard deviation (Figure 4d) shows the same gradient; however, its values were very low (<0.006 mW/cm^2^·sr·µm). The radiance response of the Spectralon to the halogen lamp is shown in Figure 4e. Dividing the spectral radiance by the DN value showed that the results were consistent to those of [40], where the spectral radiance response of the Spectralon had low values throughout most of the wavelengths. Wavelengths around 400 nm showed a slight increase in the radiance response values compared to larger neighboring wavelengths, while the maximum peak values are found close to 1000 nm. All spatial pixels showed consistent and similar measurements in both the DN and radiance domains.

No hot pixels were identified in the DN mean spectral signature. For the radiance signal, only one pixel showing values above neighboring pixels was detected in the NIR region. This pixel was one of those previously identified in the dark current experiments, corresponding to pixel 135 at wavelength 995.5 nm. Readings of this pixel were approximately 17.5 mW/cm^2^·sr·µm, while most of the neighbor pixel–waveband combinations were <16.5 mW/cm^2^·sr·µm. Results were similar for both of the white reference experiments. It can be inferred that the influence of the hot pixels identified in the dark current experiment is negligible when the sensor is measuring the intensity of light across wavebands.

Figure 5 shows the temporal variability analysis in the radiance domain. Figure 5a shows that wavelength channels below 800 nm had preferentially lower or closer to zero values by the end of the experiment compared to the first measurements. The most significant drops in radiance were on average −1.5% around 500 nm, with the largest drop of approximately 1.9% occurring for spatial pixel number 13 at 514.12 nm. Channels recording above 800 nm showed consistently higher radiance values after 30 min (Figure 5a). The largest increases of radiance were on average approximately 3.5% and occurred for wavelengths close to 1000 nm, with the largest increase being approximately 5%, identified for spatial pixel 496 at 407.06 nm. The absolute percentage of change was <4% in 99.9% of cases. In general, the camera’s response was stable for normal UAV operation times, i.e., 30 min.

For the first white reference experiments, approximately 46% of the pixel–waveband combinations showed an increase in their final radiance measurement after 30 min, with a lower reading occurring for the remaining pixels. For the white reference experiment, this corresponds to 92,489 pixel–waveband combinations showing a decrease in their last measurement. However, for the second white reference experiment, approximately 61% of the pixel–waveband combinations showed an increase in their final radiance measurement. In both experiments, wavelengths above 800 nm showed an increase in their readings throughout the spatial pixels, which means the difference between both experiments arises from the region of the spectrum below 800 nm, where the number of pixel–waveband combinations that had a larger or lower reading after 30 min varied through repeated uses of the camera. However, variations in the NIR region are much closer to zero and are likely not a concern for UAV campaigns.

Figure 5b shows the mean, maximum, and minimum percentage of variation of all pixel–waveband combinations for each minute in relation to the first measurement. Again, all the regions in the spectrum followed the shape of the global mean, but each curve varied by a different percentage. The blue and red parts of the spectrum (see individual lines in Figure 5b) showed similar responses and were consistently lower than their first reading. The same is true for the green part of the spectrum, which had a more negative relative change with respect to the first measurement than the blue and red parts of the spectrum. Only the NIR region of the spectrum showed larger readings compared to the first measurement, with a positive change occurring after 10 min, which might result in larger NIR values in the last swaths collected during a survey. However, aside from the maximum and minimum values, the relative difference throughout the experiment was between −2% and 2%, which is unlikely to affect most hyperspectral surveys. Based on these results, it can be concluded that no warm-up period is required before camera deployment due to optical effects.

### 3.2. Spectral Calibration Assessment

Experiments were performed to assess the calibration quality of the spectral wavebands using a Hg/Ar lamp (refer to Section 2.3). The Hg/Ar intensity values per wavelength as measured by the Nano-Hyperspec sensor were compared to documented Hg and Ar values, as well as those measured with the ASD Fieldspec-4 spectroradiometer (Figure 6). The vertical scale, formerly in a DN arbitrary intensity scale of each device, was normalized to allow for easier comparison. The most important aspect here was the wavelength position of the peaks. For the Fieldspec-4 spectroradiometer, the peaks were coincident and reflected the reported Hg and Ar emission wavelength lines, indicating an optimal spectral calibration (Figure 6).

The peaks of the Nano-Hyperspec data closely matched the peaks for the documented Hg/Ar emission wavelengths, although slight offsets of 1 to 3 nm were identified. The shift to the right encountered for the peaks registered by the Nano-Hyperspec sensor occurred for all the reported wavelengths, with the offset approximately the same for all wavelength peaks. Shifts in the recorded spectrum with respect to the calibration standard need to be accounted for. Otherwise, the deviations will generate low spectral fidelity of the sampled ground features. For the Nano-Hyperspec sensor, the region below 500 nm showed slightly elevated values above zero at locations with no emission lines. Both the ASD Fieldspec-4 and Nano-Hyperspec sensors registered an additional peak at 966 nm that was not included in the specifications of the Hg/Ar lamp. This peak was more evident in the Nano-Hyperspec measurements than those with the ASD Fieldspec-4. However, since it was registered by both instruments, it is likely not to be an artifact. An Argon plasma emission lines graphic by Ocean Optics [45] showed an emission line in the same 966 nm spectral region which likely corresponds to that presented here.

### 3.3. Conversion from Radiance to Reflectance

As described in Section 2.5, three types of radiometric calibration panels were used to assess the relationship between the Nano-Hyperspec derived radiance values and ASD Fieldspec-4 derived reflectance values. These included Spectralon, Masonite and oak plywood panels. A best-fit equation was calculated for all the wavelengths for each of the panels. The relationship between radiance and reflectance was found to be linear, with a near-perfect fit (R^2^ > 0.99) for all wavelengths with all panels. Variations in the R^2^ values between the three different sets of panels were negligible. Figure 7 shows the residuals for the linear regressions versus the fitted reflectance values for four wavebands corresponding to the RGB and NIR region of the spectrum. The figure also presents the corresponding R^2^ for the linear regressions. Residuals show a random distribution around zero for all three sets of panels, which is indicative of no bias in the regression models. The magnitude of the residuals changed between the different types of panels, with the Spectralon panels having the residuals closest to zero, while the oak plywood panels had the most dispersed residuals. Residuals ranged between −0.009 and 0.004 for the Spectralon panels, between −0.006 and 0.006 for the Masonite panels, and between −0.033 and 0.022 for the oak plywood panels. Residuals increased by an order of magnitude from the Spectralon and Masonite to the oak plywood panels, which indicates that the former materials are better suited as radiometric calibration panels than those of oak plywood. These results align with those presented by [32]. However, all residuals were very close to zero, indicating that oak plywood can also be used for the empirical line method.

It is essential that panels used for an empirical line method cover a wide range of spectrally homogenous and known reflectance values, which can only be attained with the aid of a spectroradiometer, such as the Fieldspec-4, in the case of user-painted panels. While the Spectralon grey scale panels provide known standards with widespread reflectance levels, Spectralon panels of appropriate size for most flight campaigns or equivalent reflectance coatings are expensive and would be largely exposed to wear and tear from field conditions that would degrade their spectral properties rapidly. The use of Masonite and/or oak plywood panels coated with matte paint [30,46] to perform the empirical line method is supported by our results, although Masonite is preferred. Their use, however, requires a spectroradiometer to measure reflectance of the constructed grey scale panels. Therefore, the panels should not replace the implementation of a Spectralon surface as a white reference for the spectroradiometer and for the Nano-Hyperspec camera during pre-flight workflows to guarantee that the calibration steps are optimized.

## 4. Discussion

UAV-based hyperspectral measurements provide a relatively new and potentially powerful means for deriving high-spatial, -spectral, and -temporal resolution information. UAV platforms are suited for bridging the scaling gap between field studies, airborne remote sensing, and even satellite remote sensing data [47]. However, for the data to be useful, confidence in the stability and repeatability of measurements, together with absolute radiometric accuracy, is required. Achieving this, demands well-calibrated instruments, which require routine and repeated camera evaluations. For all UAV-based push-broom hyperspectral sensors, we recommend that the radiometric accuracy of the data is carefully assessed using the procedures outlined in this paper, i.e., dark current and white reference measurements, evaluation of the spectral band calibration, and the relation and conversion of derived radiance values to at-surface reflectance. Without a basic understanding of radiometric accuracy, it is impossible to determine the quality of the collected data. Once noise levels have been quantified, it is possible to calculate the minimum detectable difference in reflectance values required to differentiate features being mapped, e.g., two different tree species. Similarly, it is possible to determine the minimum detectable change required to establish changes in biophysical properties and plant traits, e.g., leaf area index or leaf chlorophyll condition [48,49]. Propagating noise values through applied models for mapping, e.g., leaf area index or chlorophyll, enables a threshold to be set beyond which detected change represents actual change as opposed to change caused by noise in the image data. These types of calculations can also be related to unexplained variance in prediction models [44]. Procedures like these will enable users of hyperspectral data to determine if the radiometric accuracy of their data is suitable for a particular application, or if calibration of their sensor (or even the use of a different sensor) is necessary to achieve the required results.

Our results for the dark current assessment indicated that increasing the exposure from 6 ms to 12.5 ms increased the maximum difference between the first and last radiance measurements from 5.6% to 7.3%. However, a relative difference between the first and last measurements greater than 5% only occurred in 0.005 and 0.004% of the pixel–waveband combinations when using an exposure of 12.5 ms and 6 ms, respectively. Overall, our results showed that the average dark current levels for the Nano-Hyperspec system in each of the spectral regions in the radiance domain increases <2% after 30 min, which is well below the 5% threshold used by [32]. As only a pre-flight dark current measurement is collected and used to subtract from the recorded data during a survey, the smaller the increases in dark current are, the less impact it will have on the last swaths taken during a flight and the more stable the bands can be considered. Our outcomes showed dark current spatial inhomogeneities within the same band in the DN domain. These inhomogeneities were eliminated when processed into radiance. Regarding the radiometric response for the white reference experiments, it was shown that the bands have a uniform response across spatial pixels in both the DN and radiance domains. The results for the temporal variability of the white reference experiments indicated relative differences well below 5% even when considering its maximum values, implying that temporal changes with the lens cap off are negligible.

In our evaluation, no dead pixels were detected, but 17 pixel–waveband combinations showed values larger than 135 DN, corresponding to 0.01% of the total number of combinations. Conversion into radiance values reduced the hot pixels to one single pixel–waveband combination. Taking as a reference the 0.9% limit for dead pixels provided by camera manufacturer, Headwall Photonics, for the camera evaluated in [34], the Nano-Hyperspec can be considered to be in optimal condition regarding hot and dead pixels. Moreover, due to the very high spatial resolution of UAV-based imagery, object-based image analysis approaches are likely to be used for mapping purposes, as the features being mapped often consist of hundreds or thousands of pixels [50,51,52]. A few hot pixels would hence be averaged out for objects, consisting of a large number of pixels when undertaking the object-based image analysis as opposed to per-pixel analysis.

Assessments of spectral calibration in other studies [32,33,34] have shown shifts in peak center wavelength and increases in FWHM. Shifts in peak center wavelength ranged between 5 to 10 nm in Hakala et al. [32] and 4 to 15 nm in Arngren [34]. Our results for the Nano-Hyperspec data showed lags between 1–3 nm, which are comparatively smaller. Depending on the FWHM and the lag magnitude, this can have detrimental effects for high-precision applications [35]. Correction methods range from linear [33] or polynomial models, calibration standards [34] or during flight methods based on in situ surface reflectance [35]. A temporal assessment of the spectral fidelity to identify shifts in the spectral axis during operational time was not carried out in our work. However, it can be performed conducting two experiments with the Hg/Ar light source. First, an initial measurement can be acquired a few minutes after turning the camera on. While the camera can be left on for 30 min inside the basic setup, the Hg/Ar light source should be turned off to avoid overheating. After 30 min, the light source can be turned back on and a new measurement can be taken. This will only evaluate the influence of the internal heating of the camera on the spectral calibration. Heating during fieldwork and its effects on the spectral axis should be examined separately, taking into account that for recording the Hg/Ar lamp spectrum, the camera should be isolated from any other light sources. It is recommended for future work to assess the spectral fidelity over the temporal scale of standard UAV operations.

Following its use in many studies [53,54], the empirical line method has proved to be a standard technique for converting multispectral and hyperspectral radiance measurements to reflectance values. Experiments presented in [33,35] showed strong linearity between DN and at-sensor radiances appropriate for absolute radiometric calibration. In our experiments, we used the empirical line method to convert radiance values to reflectance, assuming conversion from DN to radiance is adequately achieved by the SpectralView software. Our results showed a near-perfect linear relationship for all bands between the Nano-Hyperspec derived radiance and ASD Fieldspec-4 measured reflectance of three different kinds of radiometric calibration panels. The use of the empirical line method may prevent the need for more sophisticated radiative transfer modeling approaches. If a spectroradiometer is available, painted Masonite or oak plywood panels can be used as low-cost alternatives for converting radiance to reflectance measurements, with Masonite being the preferred material. However, the use of Masonite or oak plywood panels should not replace good practices for recording white reference measurements with a spectroradiometer and the Nano-Hyperspec. A Spectralon panel with 100% reflectance should be used for spectral optimization and collection of white reference measurements for each survey. When using alternative panel materials, it is important to ensure that they are near-Lambertian, i.e., reflect light evenly in all directions [16,55], and have uniform reflectance [16]. They should be placed horizontally and have an area large enough to produce pure pixels according to the resolution of the images. Shading of the panels must be avoided. The illumination conditions should be as constant as possible; hence, these experiments or actual field surveys should be undertaken under clear sky conditions and only cover short time periods to avoid substantial solar angle changes [16].

The workflow followed in this document and presented in Figure 1 corresponds to our suggested assessment protocol for ensuring consistency, reliability, and repeatability of the collection of UAV-based hyperspectral push-broom data. For the relative radiometric calibration assessment experiments, it is recommended to keep the camera away from any other sources of radiation, which might increase its temperature. For the dark current experiments, the cap of the lens should be on before starting the experiment, and it is advisable to turn external lights off. The camera should also be allowed to cool to room temperature before running subsequent experiments. Each of our experiments was conducted one day apart. For the white reference experiments, the Spectralon panel should be placed horizontally, and the camera should be pointed toward the panel at nadir, while the angle of the halogen lamp must be kept constant. Ideally, the set-up should not be moved between experiments. The spectral calibration assessment experiments require no sources of radiation other than the Hg/Ar lamp. The lamp should be pointed directly to the lens, but within a safe distance from it (2 to 3 cm) to avoid touching the camera lenses. The intensity should be large enough to allow a clear capture of the lamp’s spectrum and can be attained by adjusting the exposure. The implementation of the protocol suggested in this study showed that overall, the Nano-Hyperspec camera was spatially consistent and time stable. Moreover, this protocol is suitable for any push-broom hyperspectral camera. The workflow and the experimental setup may also be used for other types of cameras. The data processing is largely the same with pixels being averaged between successive images. The end product for a snapshot camera would be a 2D array of pixels instead of the 1D array obtained with push-broom sensors, which might complicate data visualization for the wavelengths recorded. The analysis of snapshot sensors should also include an assessment of vignetting effects.

## 5. Conclusions

As with any instrument, UAV-based hyperspectral sensors require a thorough evaluation prior to their deployment. Assessing a camera’s performance ensures the quality of the data being collected and facilitates the detection of any sources of error or noise. Knowing the noise levels of recorded spectral signatures across a sensor’s spatial pixels and over time allows the quantification of how these are likely to impact mapping results and the assessment of collected reflectance values to determine their suitability to confidently monitor land-cover change, discriminate vegetation species, and assess variations in biophysical and biochemical plant properties.

Results for the relative radiometric calibration assessment performed here were consistent throughout the experiments, indicating that the electronic and optical components of the Nano-Hyperspec sensor are working in a manner that allows accurate spectral information to be collected. For the relative radiometric calibration experiments, the output can be considered uniform in space and time for both the dark current and the white reference assessments. Identified deviations or non-uniformities were reproduced for every experiment, and their influence characterized to allow removal if necessary. Overall, it was determined that no warm-up period is required prior to camera deployment due to dark current or optic effects. The spectral calibration showed a shift between 1 and 3 nm of the measured peaks of spectral intensity with respect to the emission lines of documented Hg/Ar wavelengths. This small shift was found to be consistent and can be rectified by adjustment, as previously suggested. Conversion from radiance to reflectance through the empirical line method was achieved with optimal results for all three panel materials employed. As long as the Masonite and oak plywood panels are carefully produced to ensure even and near-Lambertian reflectance properties, they can provide a convenient and cost-effective solution for estimating reflectance values. Panels can be replaced as needed when normal wear and tear caused by field conditions degrade their spectral properties.

The work presented here constitutes a simple, yet effective, protocol for assessing the performance and radiometric accuracy of UAV-based hyperspectral push-broom cameras. Future work should expand the evaluation to non-laboratory UAV-based experiments and develop suitable data quality assurance steps and protocols for assessing the impact of flight planning, acquisition parameters and external factors on the radiometric accuracy of the Nano-Hyperspec image data.

## Figures and Tables

**Figure 1 sensors-19-04699-f001:**
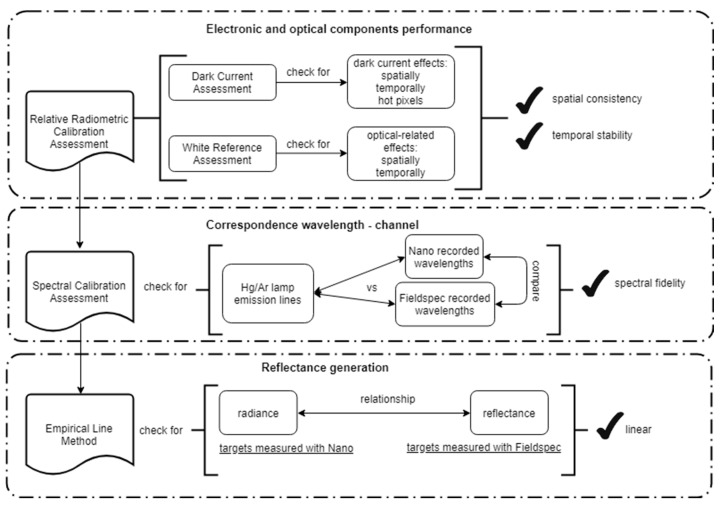
Workflow used in this study for assessing the performance of the push-broom hyperspectral sensor.

**Figure 2 sensors-19-04699-f002:**
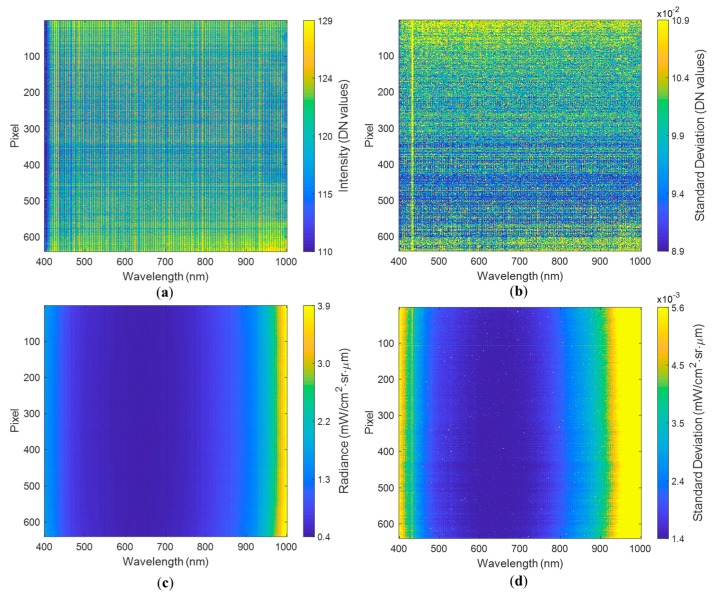
Spectral signature for the dark current (covered lens) experiment with an exposure of 6 ms, showing (**a**) DN mean spectral signature, (**b**) standard deviation of the DN spectral signature, (**c**) radiance mean spectral signature, and (**d**) standard deviation of the radiance spectral signature.

**Figure 3 sensors-19-04699-f003:**
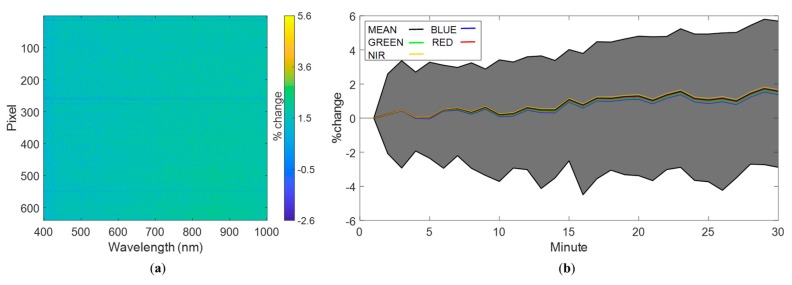
Temporal analysis for the dark current experiment with an exposure of 6 ms, showing (**a**) percentage of variation on a range from −2.5% to 5.6 between the first and the last radiance measurements, and (**b**) time series analysis of the percentage of variation from the first measurement. The gray region represents the area enclosed between the maximum and minimum values of relative difference. The global mean is depicted in black, the mean for the red-green-blue (RGB) regions of the spectrum are depicted in red, green, and blue, respectively, and the near-infrared (NIR) region is shown in yellow.

**Figure 4 sensors-19-04699-f004:**
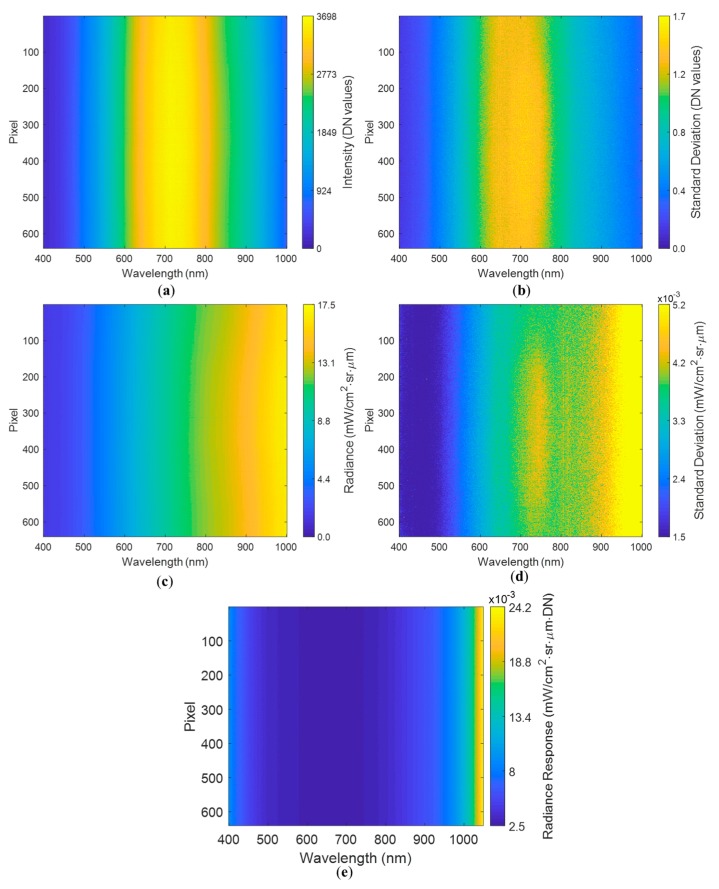
Spectral signature for the white reference with an exposure of 12.5 ms, showing (**a**) DN mean spectral signature, (**b**) standard deviation of the DN spectral signature, (**c**) radiance mean spectral signature, (**d**) standard deviation of the radiance spectral signature, and (**e**) Spectralon radiance response.

**Figure 5 sensors-19-04699-f005:**
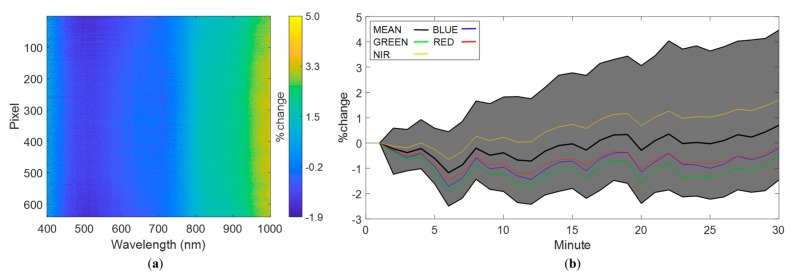
Temporal analysis for the white reference experiment at an exposure of 12.5 ms, showing (**a**) percentage of variation between the first and the last radiance measurement at 30 min, and (**b**) time series analysis of the percentage of variation in radiance from the first measurement. The gray region represents the area enclosed between the maximum and minimum values of relative difference. The global mean is depicted in black, the mean for the RGB regions of the spectrum are depicted in red, green, and blue, respectively, and the NIR region is shown in yellow.

**Figure 6 sensors-19-04699-f006:**
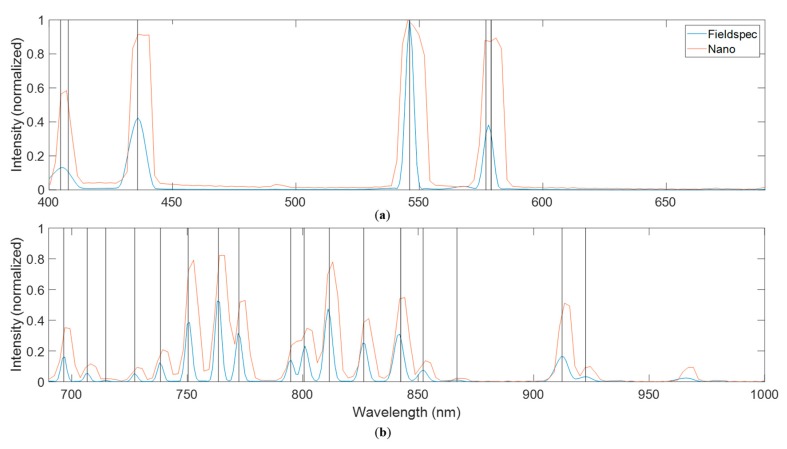
Normalized intensity comparison between Hg lamp (**a**) and Ar lamp (**b**) emission lines (vertical black lines, see Table 2 for reference values) and recorded spectra with the ASD Fieldspec-4 (blue curve) and the Nano-Hyperspec sensor (orange).

**Figure 7 sensors-19-04699-f007:**
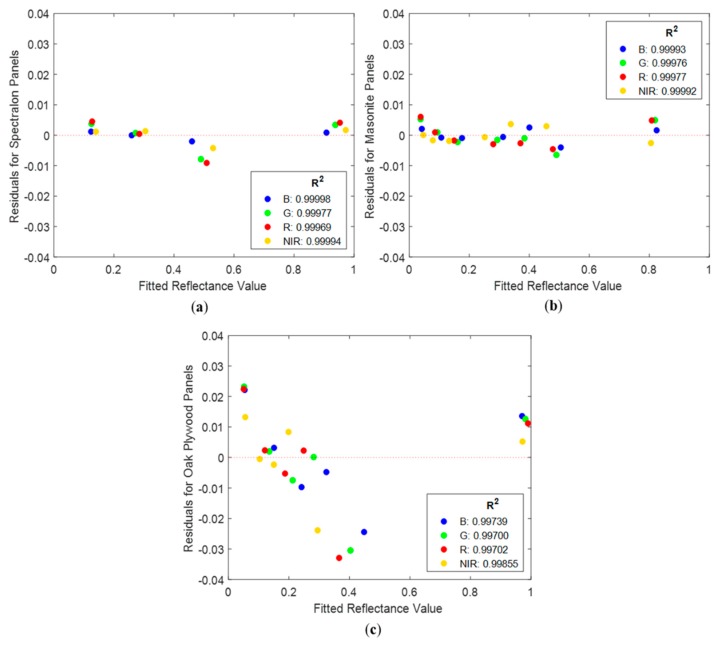
Residuals vs. fitted reflectance plots for (**a**) Spectralon, (**b**) Masonite, and (**c**) oak plywood panels. Blue dots correspond to a wavelength of 447 nm in the blue region of the spectrum, green dots to a wavelength of 554 nm in the green region, red dots to a wavelength of 652 nm in the red region, and yellow points to a wavelength of 850 nm in the NIR region. The legend is accompanied by R^2^ of the regression for the specific wavelengths.

**Table 1 sensors-19-04699-t001:** Details of the Nano-Hyperspec data collected for each of the experiments.

Type of Experiment	No. of Experiments	Exposure (ms)	Files per Experiment	Data Size (GB) per Experiment
Dark current	2	6	301	100
Dark current	2	12.5	140	46.9
White reference	2	12.5	137	46.3
Spectral calibration	1	1000	1	217
Empirical line	1	3	372	71.1

**Table 2 sensors-19-04699-t002:** Hg/Ar emission lines as reported from Ocean Optics for their Hg-1 lamp.

Hg Emission Lines (nm)	Ar Emission Lines (nm)
253.652	696.543
296.728	706.722
302.15	714.704
313.155	727.294
334.148	738.398
365.015	750.387
404.656	763.511
407.783 *	772.376
435.833	794.818
546.074 **	800.616 ***
576.96	811.531
579.066	826.452
	842.465
	852.144
	866.794
	912.297
	922.45

* This spectral line is not evident with spectrometers configured with 300 or 600 lines/mm gratings. ** Spectrometers with 1200, 1800, 2400 or 3600 lines/mm gratings have spectral lines evident at 576.96 nm and 579.07 nm. *** This spectral line is evident only with spectrometers configured with 1800, 2400, or 3600 lines/mm gratings. As specified by Ocean Optics [43].

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
