# Peer review of "Radiometric Assessment of a UAV-Based Push-Broom Hyperspectral Camera"

_sensors, 2019, doi:10.3390/s19214699_

Round 1
Reviewer 1 Report
In this paper, the authors proposed a general protocol for sensor evaluation and used the Nano-Hyperspec push-broom hyperspectral sensor to conduct a radiometric assessment. The experiment was mainly divided into the following three parts: (1) assess dark current and white reference consistency, both temporally and spatially;(2) evaluate spectral fidelity;and (3) determine the relationship between sensor-recorded radiance and spectroradiometer-derived reflectance. This article is well structured, clear and well written, but it lacks of technical contribution, since the approach is actually a syntheses of well know techniques. Other major comments are as follows:
This paper points out that this radiation quality evaluation workflow can be applied to UAV-based hyperspectral sensor and is introduced with a Nano-Hyperspec sensor. Explain whether this workflow changed when it was used for other sensors. There are no points of innovation in this paper. Please list the innovation points in this paper or the advantages over other evaluation methods. On page 8, about the selection of radiometric calibration panels, why are these three kinds of calibration panels chosen for testing? What are their differences? On page 12, line 443, please explain the radiance signature of Ref.45 and Ref.46. Lines 455- 459 on page 13, how did you get 43% and 61% of the pixel-waveband combinations? This is not mentioned in the paper.Author Response
Please see the attachment.

Reviewer 2 Report
This paper is a well organized and well conducted review of testing a hyperspectral instrument designed for UAVs. It assesses the cross track and along wavelength stability of the sensor through an expected operation time frame, as well as assessing the spectral calibration. A method to conduct this for field instruments was laid out and provides useful guidance for field deployment of this type of sensor. One piece of information/test that would have made this even more worthwhile would have been the inclusion of some measure of spectral fidelity over the temporal scale of operation. Small, uncooled hyperspectral instruments can possibly be affected by heat fluctuations affecting optics to grating to sensor spacing during operation, which can cause a shift in the spectral axis over an operational period. Even if no measurements were made that can evaluate this, it should be included in the discussion. In Line 421 - should that read spectral instead of spatial?
Round 2
Reviewer 1 Report
The authors have successfully addressed all the remarks and the paper can be accepted.